# In Vitro and In Silico Mechanistic Insights into miR-21-5p-Mediated Topoisomerase Drug Resistance in Human Colorectal Cancer Cells

**DOI:** 10.3390/biom9090467

**Published:** 2019-09-09

**Authors:** Jung-Chien Chen, Yao-Yu Hsieh, Hsiang-Ling Lo, Albert Li, Chia-Jung Chou, Pei-Ming Yang

**Affiliations:** 1PhD Program for Cancer Molecular Biology and Drug Discovery, College of Medical Science and Technology, Taipei Medical University and Academia Sinica, Taipei 111, Taiwan; 2Department of Surgery, Min-Sheng General Hospital, Taoyuan 168, Taiwan; 3Central Clinic and Hospital, Taipei 106, Taiwan; 4Division of Hematology and Oncology, Shuang Ho Hospital, Taipei Medical University, New Taipei City 235, Taiwan; 5Division of Hematology and Oncology, Department of Internal Medicine, School of Medicine, College of Medicine, Taipei Medical University, Taipei 111, Taiwan; 6Graduate Institute of Cancer Biology and Drug Discovery, College of Medical Science and Technology, Taipei Medical University, Taipei 111, Taiwan; 7School of Medicine, Taipei Medical University, Taipei 111, Taiwan; 8TMU Research Center of Cancer Translational Medicine, Taipei 111, Taiwan; 9Cancer Center, Wan Fang Hospital, Taipei Medical University, Taipei 111, Taiwan

**Keywords:** autophagy, colorectal cancer, Connectivity Map, drug resistance, microRNA

## Abstract

Although chemotherapy for treating colorectal cancer has had some success, drug resistance and metastasis remain the major causes of death for colorectal cancer patients. MicroRNA-21-5p (hereafter denoted as miR-21) is one of the most abundant miRNAs in human colorectal cancer. A Kaplan–Meier survival analysis found a negative prognostic correlation of miR-21 and metastasis-free survival in colorectal cancer patients (The Cancer Genome Atlas Colon Adenocarcinoma/TCGA-COAD cohort). To explore the role of miR-21 overexpression in drug resistance, a stable miR-21-overexpressing clone in a human DLD-1 colorectal cancer cell line was established. The 3-(4,5-dimethylthiazol-2-yl)-2,5-diphenyl tetrazolium bromide (MTT) cell viability assay found that miR-21 overexpression induced drug resistance to topoisomerase inhibitors (SN-38, doxorubicin, and etoposide/VP-16). Mechanistically, we showed that miR-21 overexpression reduced VP-16-induced apoptosis and concomitantly enhanced pro-survival autophagic flux without the alteration of topoisomerase expression and activity. Bioinformatics analyses suggested that miR-21 overexpression induced genetic reprogramming that mimicked the gene signature of topoisomerase inhibitors and downregulated genes related to the proteasome pathway. Taken together, our results provide a novel insight into the role of miR-21 in the development of drug resistance in colorectal cancer.

## 1. Introduction

According to the global cancer statistics in 2018, colorectal cancer (CRC) is still the second essential contributor of cancer-related deaths in males and females worldwide [1]. Chemotherapeutics based on 5-fluorouracil (5-FU), in combination with oxaliplatin or irinotecan, are the standard regimen for CRC, which has increased the overall survival of CRC patients [2]. In addition, several molecular-targeted therapies have been developed, such as cetuximab and bevacizumab, which are monoclonal antibodies specific to epidermal growth factor receptor (EGFR) and vascular endothelial growth factor (VEGFR), respectively [2]. However, the acquisition of anticancer drug resistance remains the major challenge in treating CRC. A better characterization of the molecular mechanisms that are involved will provide clinical benefits for controlling and preventing the development of drug resistance.

MicroRNAs (miRNAs), approximately 19 to 25 nucleotide-long small noncoding RNAs, could suppress mRNA translation or stability through interacting with the 3’-untranslated region (3’-UTR) of target mRNAs. The alteration of miRNA expressions is frequently observed in human cancers, and these altered miRNAs have been shown to display either oncogenic or tumor-suppressive functions [3]. An oncogenic miRNA, miR-21, is overexpressed in CRC [4,5]. Tumor suppressor genes, such as programmed cell death 4 (PDCD4), phosphatase and tensin homolog (PTEN), and sprouty 2 (SPRY2), have been known as mRNA targets of miR-21. Silencing these genes stimulates cell proliferation, inhibits apoptosis, and enhances invasion and metastasis [6,7]. However, the role of miR-21 in drug resistance in CRC, and the participated molecular mechanisms remains unclear.

Here, we established an miR-21-overexpressing DLD-1 human CRC cell line to investigate its role in drug resistance in CRC. We demonstrated that miR-21 overexpression induced drug resistance to topoisomerase inhibitors through reducing apoptotic induction and enhancing autophagic flux. Bioinformatics analyses further provided in silico mechanistic insights into miR-21-mediated drug resistance. Our results suggest a novel role of miR-21 in drug resistance in CRC.

## 2. Materials and Methods

### 2.1. Chemicals and Reagents

Lipofectamine RNAiMAX Transfection Reagent, Roswell Park Memorial Institute-1640 (RPMI-1640) medium, sodium pyruvate, L-glutamine, and antibiotic/antimycotic solution were from Life Technologies (Gaithersburg, MD, USA). Fetal bovine serum (FBS) was from GIBCO (Grand Island, NY, USA). Programmed cell death 4 (PDCD4; GTX104901), DNA topoisomerase I (TOP1; GTX63013), DNA topoisomerase II alpha (TOP2A; GTX100689), DNA topoisomerase II beta (TOP2B; GTX102640), Sequestosome 1 (SQSTM1/p62; GTX100685), Microtubule associated protein 1 light chain 3 beta (MAP1LC3B; GTX127375), Beclin 1 (GTX113039), Autophagy related 12 (ATG12; GTX124181), Lysosomal associated membrane protein 2 (LAMP2; GTX103214), and Glyceraldehyde-3-phosphate dehydrogenase (GAPDH; GTX100118) antibodies were from GeneTex (Hsinchu, Taiwan). The caspase-3 (3004-100) antibody was from Imgenex (San Diego, CA, USA). The autophagy related 7 (ATG7; sc-8668) antibody was from Santa Cruz Biotechnology (Santa Cruz, CA, USA). The ubiquitin (MAB1510) antibody was from EMD Millipore (Billerica, MA, USA). The poly(ADP-ribose) polymerase 1 (PARP1; 9542S) antibody was from Cell Signaling Technology (Beverly, MA, USA). Horseradish peroxidase (HRP)-labeled secondary antibodies were from Jackson ImmunoResearch (West Grove, PA, USA). pCMV-MIR and pCMV-MIR21 plasmids were from OriGene (Rockville, MD, USA). ON-TARGETplus human ATG7 and Non-Targeting SMARTpool siRNAs were from Dharmacon (Lafayette, CO, USA). The PolyJet DNA Transfection Reagent was from SignaGen Laboratories (Ijamsville, MD, USA). Geneticin (G418) was from Invivogen (San Diego, CA, USA). The GENEzol TriRNA Pure Kit was from Geneaid Biotech (New Taipei City, Taiwan). The miScript II RT Kit, miScript SYBR Green PCR Kit, and Hs_miR-21_2 (hsa-miR-21–5p) miScript Primer Assay were from Qiagen (Valencia, CA, USA). The iScript cDNA Synthesis Kit was from Bio-Rad Laboratories (Richmond, CA, USA). Dimethyl sulfoxide (DMSO), MG132, and 3-(4,5-dimethylthiazol-2-yl)-2,5-diphenyl tetrazolium bromide (MTT) were from Sigma Chemical (St. Louis, MO, USA). Bafilomycin A1 and doxorubicin were from LC Laboratories (Woburn, MA, USA). 5-Fluorouracil (5-FU) was from Merck Millipore (Billerica, MA, USA). Etoposide (VP-16) and 7-ethyl-10-hydroxy-camptothecin (SN-38) were from Adooq BioScience (Irvine, CA, USA). Rapamycin was from Cayman Chemical (Ann Arbor, MI, USA). An enhanced chemiluminescence (ECL) system was from Perkin-Elmer (Boston, MA, USA). Phosphatase and protease inhibitor cocktails were from Roche (Indianapolis, IN, USA). Hybond-C Extra nitrocellulose membranes were from GE Healthcare (Piscataway, NJ, USA). X-ray film and 2x SYBR Green PCR Master Mix were from Roche (Indianapolis, IN, USA).

### 2.2. Cell Culture and Transfection

DLD-1 human colorectal cancer cells were cultivated in RPMI-1640 medium containing 10% FBS, 1% L-glutamine, 1 mM sodium pyruvate, and 1% antibiotic/antimycotic solution. Cells were incubated in a humidified 37 °C, 5% CO_2_ incubator. To establish stable miR-21-overexpressing DLD-1 (DLD-1-miR-21) and corresponding vector-overexpressing (DLD-1-vector) cell lines, parental DLD-1 cells were transfected with a plasmid encoding human miR-21 or its control vector (pCMV-MIR) using the PolyJet transfection reagent. Stable clones were selected with 1 mg/mL G418 for 2 months. Single clones were selected based on the expression of the green fluorescent protein (GFP) reporter and then pooled together for subsequent experiments. For transient ATG7 knockdown analysis, human ATG7 and control scramble siRNAs were transfected into cells by Lipofectamine RNAiMAX Transfection Reagent. After 24 to 48 h, the transfected cells were used for further experiments.

### 2.3. Determination of Cell Proliferation and Cell Viability

To determine cell proliferation, the cells (10^5^) were spread in 60-mm dishes and cultured for approximately 1 to 4 days (three dishes per time point). Then, cells were harvested by trypsinization, and the cell number was counted with a hemocytometer (Marienfeld, Lauda-Königshofen, Germany). For MTT cell viability, cells were spread in 96-well plates and exposed to drugs for 72 h (five wells per treatment). At the end of drug incubation, MTT (0.5 mg/mL) was directly added to cells and cultured for an additional 4 h. Then, the medium was removed, and DMSO was added to dissolve the blue MTT formazan precipitates. Cell viability was determined based on the absorbance at 570 nm.

### 2.4. Quantitative Real-time Polymerase Chain Reaction (qPCR)

Total RNA was isolated with a GENEzol TriRNA Pure Kit. To determine the miR-21 expression, first-strand complementary (c) DNA was synthesized using a miScript II RT Kit, and then qPCR was performed using a miScript SYBR Green PCR Kit and Hs_miR-21_2 miScript Primer assays. To determine mRNA expression, first-strand cDNA was synthesized using an iScript cDNA Synthesis Kit, and then the qPCR was performed using 2x SYBR Green PCR Master Mix. The primers specific to human PDCD4 and β-actin were as follows: 5′-TGGATTAACTGTGCCAACCA-3′ and 5′-TCTCAAATGCCCTTTCATCC-3′ (PDCD4); 5′-GTTGCTATCCAGGCTGTGCT-3′ and 5′-AGGGCAT ACCCCTCGTAGAT-3′ (β-actin). Each assay was performed in triplicate on a LightCycler Nano Real-Time PCR System (Roche). The related gene expression was calculated by the comparative CT method.

### 2.5. Western Blot Analysis

Cells were lysed on ice for 30 min in ice-cold lysis buffer (50 mM Tris-HCl (pH 7.5), 2 mM EDTA, 150 mM NaCl, 1 mM MgCl_2_, 1% NP-40, 1 mM DTT, 10% glycerol, 1× protease, and phosphatase inhibitor cocktails). Protein lysates were separated on an SDS-polyacrylamide gel (SDS-PAGE) and then transferred onto a nitrocellulose membrane. After pre-hybridization in 5% skim milk/TBST (20 mM Tris-HCl (pH 7.5), 150 mM NaCl, and 0.05% Tween-20) for 1 h, the membrane was hybridized with a primary antibody overnight at 4 °C in 1% bovine serum albumin (BSA)/TBST. After washing with TBST buffer, the membrane was hybridized with an HRP-conjugated secondary antibody for 1 h in 1% BSA/TBST. Then, the membrane was washed with TBST and rinsed with TBS. The chemiluminescence was developed with an ECL system and exposed to X-ray film.

### 2.6. Band-depletion Assay

Cells in 60-mm dishes were treated with VP-16 for 1 h, and immediately lysed with 150 μL of 1× SDS sample buffer. After vortexing and boiling at 100 °C for 5 min, protein was separated by 7.5% SDS-PAGE, and then Western blot analysis was performed.

### 2.7. Microarray, Connectivity Map (CMap), and Kyoto Encyclopedia of Genes and Genomes (KEGG) Pathway Enrichment Analyses

Total RNA was purified with the GENEzol TriRNA Pure Kit. mRNA profiles were examined using the Human OneArray Plus (Phalanx Biotech, Hsinchu, Taiwan). The raw data were deposited in the National Center for Biotechnology Information Gene Expression Omnibus (NCBI GEO) database (GSE136665). Differentially expressed genes (DEGs) were prepared according to the criteria: log2 ratio ≥1 or ≤−1; *p* value < 0.05. The full DEG list was shown in File S1. For the next-generation CMap analysis, the 193 most upregulated and 167 downregulated genes (Appendix A) were input to the CLUE (https://clue.io/) [8] database to obtain 150 upregulated and 150 downregulated valid genes for querying (SEP 06, 2019, date last accessed). For the KEGG pathway enrichment analysis, the best (smallest *p* value) 3000 DEGs were prepared to run a Gene Set Enrichment Analysis (GSEA) against the canonical pathway database.

### 2.8. CellMinerCDB Analysis

The CellMinerCDB (https://discover.nci.nih.gov/cellminercdb/) is an interactive web-based portal for querying the relationship between genomic and pharmacological data from large-scale cancer cell lines [9]. For the correlation between TOP2B mRNA expression and VP-16 drug activity, both the “X- and Y-Axis Cell Line Sets” were set to “CTRP”. Cancer Therapeutics Response Portal (CTRP; https://portals.broadinstitute.org/ctrp.v2.1/) is a database linking genetic, lineage, and other cellular features of cancer cell lines to small-molecule sensitivity [10,11,12]. The “X-Axis Data Type” and “Y-Axis Data Type” were set to “exp: mRNA Expression (log2)” and “act: Drug Activity (AUC)”, respectively. The identifiers for X- and Y-axis were set to “TOP2B” and “etoposide”, respectively. For the correlation between miR-21 expression and VP-16 drug activity, the cell line set and data type of the X-axis were set to “CCLE” and “mir: MicroRNA”, respectively. The identifier for miR-21 was set to “hsa-miR-21”.

### 2.9. Kaplan–Meier Survival Analysis

The prognostic impact of miR-21 in CRC was analyzed using the PROGmiR database (http://www.compbio.iupui.edu/progmir) [13]. The input “has-mir-21” was queried, and Kaplan–Meier survival plots were generated based on the expression data of colon adenocarcinoma (COAD) from The Cancer Genome Atlas (TCGA; https://tcga-data.nci.nih.gov/tcga). Patients with high (*n* = 181) and low (*n* = 180) miR-21 expression were bifurcated at the median value. The prognostic impact of the miR-21-downregulated proteasome gene signature (PSME3, PSMA1, PSMA2, PSMA3, PSMB2, PSMB3, PSMB4, PSMB5, PSMB8, PSMB10, PSMC2, PSMC3, PSMC4, PSMC5, PSMC6, PSMD3, PSMD4, PSMD11, PSME1, PSMF1, and PSMD6) in CRC was evaluated using the PROGgeneV2 database (http://www.compbio.iupui.edu/proggene/) [14]. Two CRC patient datasets (GSE28722 and GSE28814) [15] were employed to generate the Kaplan–Meier survival plots. Patients with high (*n* = 61 or 63) and low (*n* = 61 or 62) miR-21 expression in GSE28722 and GSE28144, respectively, were bifurcated at the median value.

## 3. Results

### 3.1. miR-21 Overexpression is Correlated with Drug Resistance to Topoisomerase Inhibitors

To understand the prognostic implications of miR-21 in the CRC, the PROGmiR tool (http://www.compbio.iupui.edu/progmir) [13] was employed to query the term “has-mir-21” based on the expression data of colon adenocarcinoma (COAD) from The Cancer Genome Atlas (TCGA; https://tcga-data.nci.nih.gov/tcga). Kaplan–Meier survival plots for overall and metastatic-free survival in patients with high and low miR-21 expression were generated. The overall survival did not significantly differ between miR-21-high-expressing and miR-21-low-expressing patients (Figure 1A). However, patients with higher miR-21 expression had poor metastasis-free survival (*p* < 0.05), suggesting the involvement of miR-21 in CRC metastasis (Figure 1B). To investigate how miR-21 contributes to the metastasis of the CRC, a stable miR-21-overexpressing DLD-1 human CRC cell line (DLD-1-miR-21) and a corresponding vector-overexpressing cell line (DLD-1-vector) were established. The overexpression of miR-21 was authenticated by a quantitative real-time polymerase chain reaction (qPCR) (Figure 2A). The expression of PDCD4 messenger (m)RNA and protein, a well-known miR-21 target [16], was downregulated in DLD-1-miR-21 cells (Figure 2A,B). Similar growth rates were observed in DLD-1-vector and DLD-1-miR-21 cells (Figure 2C). However, bright-field microscopy observation found that DLD-1-miR-21 cells tend to gather together in monolayer culture (Figure 2D).

FOLFIRI is a standard chemotherapy for CRC, which consists of the following drugs: leucovorin (a vitamin B derivative), 5-FU (a thymidylate synthase inhibitor), and irinotecan (a topoisomerase I inhibitor). The development of anticancer drug resistance is a biological characteristic of metastatic CRC [17]. To investigate whether miR-21 overexpression contributes to the drug resistance of CRC, DLD-1-vector and DLD-1-miR-21 cells were challenged with 5-FU, SN-38 (the active metabolite of irinotecan), and topoisomerase II inhibitors, including doxorubicin and etoposide (VP-16); then, an MTT cell viability assay was performed. The results (Figure 2E) showed that both DLD-1-vector and DLD-1-miR-21 cells exhibited similar sensitivities to 5-FU treatment. However, DLD-1-miR-21 cells were significantly more susceptible to treatment with SN-38, doxorubicin, and VP-16. Therefore, DLD-1-miR-21 cells were specifically resistant to topoisomerase inhibitors.

To confirm the role of miR-21 in topoisomerase drug resistance, the drug sensitivity of Cancer Cell Line Encyclopedia (CCLE) [18,19] cell lines to VP-16 and doxorubicin were analyzed using the CellMinerCDB (https://discover.nci.nih.gov/cellminercdb/), which is a web-based resource for integrating pharmacological and genomic analyses of cancer cell lines [9]. As shown in Appendix A, the drug activity of VP-16 and doxorubicin was proportional to the gene expression of their target TOP2B, but inversely proportional to miR-21 gene expression. Therefore, cancer cells with higher miR-21 respond poorer to topoisomerase inhibitors.

### 3.2. miR-21 Overexpression Attenuates VP-16-Induced Apoptosis without Affecting Expressions or Activities of Topoisomerases

To investigate the mechanism of drug resistance to topoisomerase inhibitors by miR-21, whether drug treatment inhibited miR-21 expression was first examined. Since VP-16 exhibited the most differential effect on the cell viability of DLD-1-vector and DLD-1-miR21 cells (Figure 2E), we selected VP-16 as a representative for further investigation. As shown in Appendix A, VP-16 reduced the miR-21 expression level to 35% and 37% in DLD-1-vector and DLD-1-miR-21 cells, respectively. Thus, the inhibition of miR-21 may not be responsible for the differential effects of VP-16 in these two cell lines. Cell apoptosis was further examined by cleaving poly(ADP-ribose) polymerase 1 (PARP1) and caspase-3. PARP1 is a 116-kDa enzyme that is cleaved into 89-kDa and 24-kDa fragments during apoptosis [20]. As shown in Figure 2F, higher levels of cleaved PARP1 and caspase-3 were observed in VP-16-treated DLD-1-vector cells compared to DLD-1-miR-21 cells. Therefore, miR-21 overexpression induced drug resistance to VP-16 through reducing drug-induced apoptosis.

Since both the elevated and reduced expression of topoisomerases are reported to be associated with resistance to topoisomerase inhibitors [21,22], it is possible that miR-21 overexpression results in the downregulation of topoisomerases, leading to drug resistance. To test this possibility, expressions of topoisomerase I (TOP1) and II (TOP2A and TOP2B) were examined by Western blot analysis. However, the expression of TOP1, TOP2A, and TOP2B proteins were similar between DLD-1-vector and DLD-1-miR-21 cells (Figure 2G). Furthermore, a band-depletion assay was used to measure topoisomerase activity by detecting catalytic topoisomerase-DNA cleavage complexes trapped by topoisomerase inhibitors. During sodium dodecylsulfate (SDS)-polyacrylamide gel electrophoresis (PAGE), topoisomerase–DNA complexes have lower mobility than free enzymes [23]. As shown in Figure 2H, VP-16 depleted TOP2B in both DLD-1-vector and DLD-1-miR-21 cells, suggesting that miR-21 overexpression did not alter topoisomerase activity. Therefore, miR-21 overexpression leads to drug resistance and VP-16 without affecting the expressions or activities of topoisomerases.

### 3.3. Accelerated Autophagic flux is Associated with miR-21-Induced Drug Resistance

Autophagy (autophagic cell death) has been categorized as type II programmed cell death [24]. However, autophagy can also act as a stress-adaptation process that avoids cell death [25]. To investigate whether VP-16 also induces autophagy, the expression of two autophagy markers (LC3-II and p62) [26] was examined. LC3-I is modified by adding a phosphatidylethanolamine (PE) and converted into LC3-II, and p62 is degraded during the autophagic process [26]. As shown in Figure 3A, VP-16 did not significantly induce autophagy in either DLD-1-vector or DLD-1-miR-21 cells. Interestingly, treatment with VP-16 for 48 h blocked the conversion of LC3-I to LC3-II in DLD-1-vector cells, although the corresponding level of p62 was reduced. However, the basal level of LC3-II was higher in DLD-1-miR-21 cells, suggesting that miR-21 overexpression accelerated autophagic flux. Indeed, VP-16 induced more LC3-II accumulation at 24 h and enhanced p62 and LC3-II degradation at 48 h. Therefore, autophagy exhibits a protective role in response to VP-16 treatment, which was amplified by miR-21 overexpression.

To investigate the role of autophagy in the anticancer effect of VP-16, an autophagy inducer, rapamycin—which is a mammalian target of rapamycin (mTOR) inhibitor—was used to enhance autophagic flux. As shown in Figure 3B, rapamycin inhibited the VP-16-induced cleavage of PARP1 and caspase-3 in both cell types, suggesting that autophagy plays a cytoprotective role. To confirm this phenomenon, cell viability was evaluated by an MTT assay. Consistently, rapamycin protected both DLD-1-vector and DLD-1-miR-21 cells from VP-16’s cytotoxicity (Figure 3C). In contrast, the blockade of autophagosome–lysosome fusion by bafilomycin A1 (a vacuolar-type H^+^-ATPase inhibitor) induced more cell-killing in DLD-1-miR-21 cells (Figure 3D). To further ascertain the role of autophagy, ATG7 expression was knocked down by siRNA to inhibit autophagy, and the enhancement of VP-16-induced apoptosis in both DLD-1-vector and DLD-1-miR-21 cells was found (Figure 3E). Therefore, miR-21 overexpression contributes to drug resistance through accelerating autophagic flux, which did not result from the different expression levels of autophagy-related proteins between two cells (Figure 3F).

### 3.4. In Silico Analyses Reveal the Mechanistic Role of miR-21 Overexpression in the Development of Drug Resistance

Since one miRNA may regulate multiple target genes and other genes indirectly through modulating the corresponding transcription factors, we employed bioinformatics approaches to gain in silico mechanistic insights into the role of miR-21 in drug resistance. The Connectivity Map (CMap) is a web-based database consisting of gene expression signatures from small molecule-treated human cancer cells. By mining and comparing the queried and existing gene signatures, one can find connections among genetic knockdown/overexpression or small molecules with similar action mechanisms [8,27,28]. A microarray analysis of DLD-1-miR-21 and DLD-1-vector cells was performed, and differentially expressed genes (DEGs; Appendix A) were prepared to query the next-generation CMap database, CLUE (https://clue.io/) [8]. As shown in Figure 4A, drugs or the knockdown/overexpression of genes were classified as perturbational classes according to their functions. The top 10 most similar perturbational classes to miR-21 overexpression are shown. Interestingly, the gene expression profile of miR-21-overexpressing DLD-1 cells was similar to that of topoisomerase inhibitors, with a median score of 90.72. In particular, the average score of the HT-29 human CRC cell line treated with topoisomerase inhibitors was 97.26. Therefore, we propose that miR-21 overexpression mimics the gene signature of topoisomerase inhibitors and escapes from a drug-induced cytotoxicity, leading to drug resistance in the CRC.

The CMap analysis also indicated that miR-21 overexpression mimicked “proteasome pathway loss-of-function (LOF)” and “proteasome inhibitor” (Figure 4A). The ubiquitin–proteasome and autophagy–lysosome pathways, two major intracellular protein-degradation systems, usually act in a coordinated and complementary manner [29]. For example, the inhibition of proteasome activity was shown to induce autophagy [30]. Therefore, we propose that accelerating autophagic flux might be due to the inhibition of the ubiquitin–proteasome protein degradation system by miR-21. In support of this notion, a Kyoto Encyclopedia of Genes and Genomes (KEGG) [31,32,33] pathway enrichment analysis found that the proteasome pathway was enriched in miR-21-overexpressing DLD-1 cells (Figure 4B). Genes encoding core and regulatory particles of the proteasome system were downregulated by miR-21 overexpression (Figure 5, Appendix A). The inhibition of a proteasome pathway will lead to the accumulation of ubiquitin-conjugated proteins. Indeed, higher levels of ubiquitinated proteins were found in DLD-1-miR-21 cells with or without a proteasome inhibitor, MG132 (Figure 4B, the embedded figure).

To investigate whether the downregulation of genes related to the proteasome system reflected the prognostic impact of miR-21 overexpression in CRC, Kaplan–Meier survival plots were generated using the PROGgeneV2 prognostic database [14]. Two CRC patient datasets (GSE28722 and GSE28814) [15] with values of overall and metastasis-free survival were selected. Upregulated genes (PSME3, PSMA1, PSMA2, PSMA3, PSMB2, PSMB3, PSMB4, PSMB5, PSMB8, PSMB10, PSMC2, PSMC3, PSMC4, PSMC5, PSMC6, PSMD3, PSMD4, PSMD11, PSME1, PSMF1, and PSMD6) of the proteasome system (Appendix A) were used as a combined gene signature for the miR-21-mediated downregulation of the proteasome pathway. Similar to the prognostic implication of miR-21 (Figure 1), only metastasis-free survival was predicted by the gene signature in CRC patients (Figure 6). CRC patients with a lower expression of the gene signature had poor metastasis-free survival (Figure 6B), which was opposite to the prognostic impact of miR-21 (Figure 1B).

## 4. Discussion

Approximately 25–50% of CRC patients develop metastatic disease, in which liver metastasis (about 20%) is the most common distant metastatic site [34,35]. Once metastasis has occurred, it is unlikely that the disease can be completely cured due to the development of drug resistance [17]. miR-21 is the most abundant miRNA in CRC, and is associated with tumor proliferation, invasion, and metastasis [36,37,38]. In this study, we found that CRC patients expressing high levels of miR-21-high had poor metastasis-free survival. In addition, miR-21 overexpression in CRC cells resulted in the development of topoisomerase inhibitor resistance. Therefore, a better characterization of the underlying molecular mechanisms of drug resistance induced by miR-21 overexpression will provide clinical benefits to control or prevent metastatic disease.

As an oncogenic miRNA, miR-21 is well-known to mediate resistance to radiation and chemotherapy in various cancer types [39,40,41]. However, its role in CRC drug resistance is less clear. 5-FU-based chemotherapy in combination with oxaliplatin (such as FOLFOX) or irinotecan (such as FOLFIRI) is the standard regimen for CRC [2]. The upregulation of miR-21 has been found in drug-resistant CRC cell lines, including the FOLFOX (5-FU/leucovorin/oxaliplatin)-resistant HCT-116 and HT-29 cells [42], 5-FU-resistant DLD-1 and KM12C cells [43], oxaliplatin-resistant HCT-15 and SW480 cells [44], and doxorubicin-resistant HT-29 and LoVo cells [45]. miR-21-mediated FOLFOX resistance is associated with the upregulation of cancer stemness by inhibiting transforming growth factor beta receptor 2 (TGFβR2) and activating the Wnt/β-catenin pathway [42]. How miR-21 mediates the resistance of other drugs, such as oxaliplatin and doxorubicin, has not been characterized. In our study, we found that stable miR-21 overexpression led to the resistance to topoisomerase inhibitors (SN-38, VP-16, and doxorubicin) in DLD-1 cells. Mechanistically, miR-21 overexpression reduced VP-16-induced apoptosis and enhanced VP-16-induced autophagic flux. Whether SN-38 and doxorubicin acted in similar ways warrants further investigation. In contrast, miR-21 overexpression did not influence the sensitivity of DLD-1 cells to 5-FU. It has been found that mismatch repair (MMR)-proficient CRC cells respond better to 5-FU therapy [46,47]. In addition, miR-21 overexpression induces 5-FU resistance by downregulating the core MMR recognition protein complex, human mutS homolog 2 (hMSH2) and 6 (hMSH6) in CRC [48]. DLD-1 cells were defective in the hMSH6 gene [49], which may partly explain the inconsistent role of miR-21 in 5-FU resistance.

Recently, the role of autophagy in topoisomerase inhibitor resistance has received attention in this field [50]. Since autophagy plays dual functions in cancer, the modulation of autophagy, either by reversing cytoprotective autophagy or by promoting cytotoxic autophagy, could overcome cancer resistance to topoisomerase inhibitors [50,51,52]. Our results indicate that miR-21 overexpression stimulated autophagic flux, which antagonized topoisomerase inhibitor-induced apoptosis. The inhibition of autophagic flux by bafilomycin A1 or ATG7 knockdown thereby selectively killed DLD-1-miR-21 cells or potentiated VP-16-induced apoptosis, respectively. In contrast, the enhancement of autophagic flux by rapamycin exacerbated miR-21-mediated drug resistance.

The role of miR-21 in regulating autophagy is still controversial. miR-21 inhibits autophagy through inhibiting its target, PTEN, and the subsequent activation of AKT [53,54]. However, the activation of ERK by miR-21 promotes arsenite-induced autophagy [55]. In addition, miR-21 enhances autophagic flux through promoting lysosomal function [56]. Our results found that miR-21 may disrupt proteasome function through downregulating the related gene expression. As the compensative behavior between the proteasome and autophagy [30], miR-21-mediated impairment of the proteasome may lead to the overactivation of autophagy.

Our results implied that the clinical efficacy of chemotherapy containing topoisomerase inhibitors for CRC patients would be obstructed by miR-21 overexpression. However, it is argued that the dosages of anticancer drugs used for in vitro experiments are usually higher than those that can be achieved in clinical settings, and the effects observed in vitro may result from the clinically irrelevant off-target activities. It has been suggested that the maximum plasma concentration (*C*max) of a drug in clinic could be a parameter being reverse-translated to in vitro experimental designs [57]. The reported *C*max values for SN-38, doxorubicin, and VP-16 are 0.143 μM, 6.73 μM, and 33.4 μM, respectively [57,58,59,60]. Therefore, the dosages of these drugs used in this study are clinically relevant.

## 5. Conclusions

In conclusion, we found that miR-21 overexpression induced the drug resistance of CRC cells to topoisomerase inhibitors through inhibiting apoptosis and enhancing autophagic flux. Bioinformatics analyses further provided in silico mechanistic insights into miR-21-mediated drug resistance. First, miR-21 overexpression mimics the gene signature of topoisomerase inhibitors, which may lead to the reduced responsiveness of CRC cells to drug treatment. Second, miR-21 overexpression inhibits the proteasome pathway through downregulating gene expression, which may lead to the enhancement of autophagic flux. Since the relationship between miR-21 and the proteasome pathway has never been investigated, further exploration is needed.

## Figures and Tables

**Figure 1 biomolecules-09-00467-f001:**
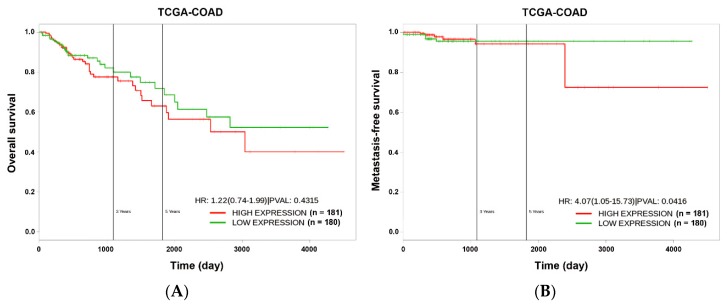
The prognostic impact of microRNA-21-5p (miR-21) on overall (**A**) and metastasis-free (**B**) survival in colorectal cancer (CRC). The prognostic impact of miR-21 in CRC was analyzed using the PROGmiR database (http://www.compbio.iupui.edu/progmir). The input “has-mir-21” was queried, and Kaplan–Meier survival plots were generated based on the expression data of colon adenocarcinoma (COAD) from The Cancer Genome Atlas (TCGA; https://tcga-data.nci.nih.gov/tcga). Patients with high (*n* = 181) and low (*n* = 180) miR-21 expression were bifurcated at the median value. HR, hazard ratio; PVAL, *p* value.

**Figure 2 biomolecules-09-00467-f002:**
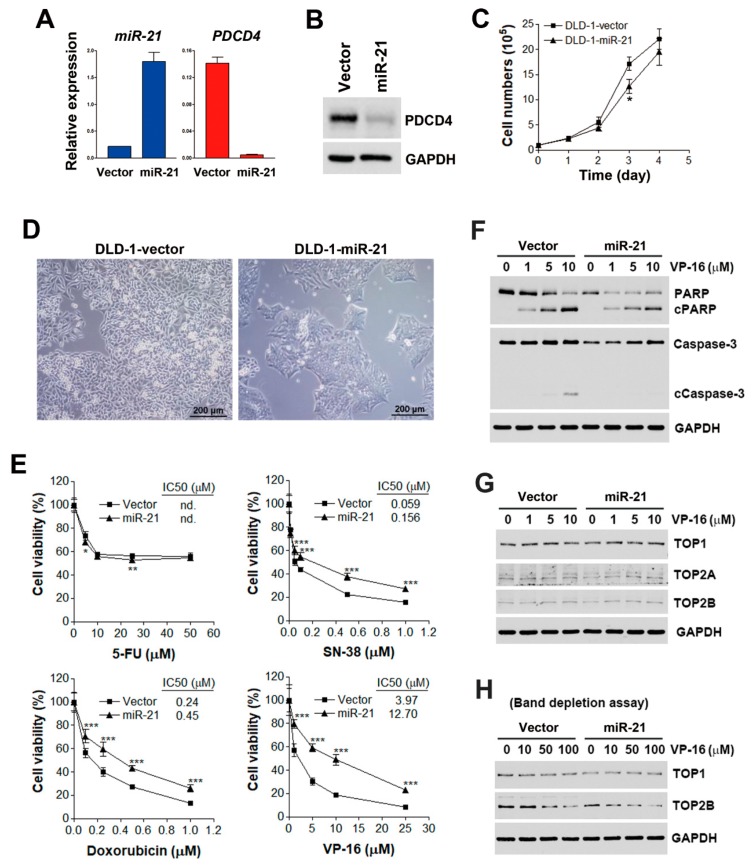
Effect of miR-21 overexpression on chemosensitivity. (**A**) mRNA expression of miR-21 and programmed cell death 4 (PDCD4) in corresponding vector-overexpressing DLD-1 (DLD-1-vector) and miR-21-overexpressing DLD-1 (DLD-1-miR-21) cells were analyzed by qPCR. (**B**) Protein expressions of PDCD4 in DLD-1-vector and DLD-1-miR-21 cells were analyzed by Western blot analysis. (**C**) Growth rates of DLD-1-vector and DLD-1-miR-21 cells were measured by cell counts at approximately 1 to 4 days. *p* < 0.05 (*) indicates significant differences between DLD-1-miR-21 and DLD-1-vector cells. (**D**) Cell morphology was observed under bright-field microscopy. (**E**) DLD-1-vector and DLD-1-miR-21 cells were treated with various doses of 5-fluorouracil (5-FU), SN-38, doxorubicin, and VP-16 for 72 h. Cell viability was analyzed by an 3-(4,5-dimethylthiazol-2-yl)-2,5-diphenyl tetrazolium bromide (MTT) assay. *p* < 0.05 (*), *p* < 0.01 (**), or *p* < 0.001 (***) indicates significant differences between DLD-1-miR-21 and DLD-1-vector cells. n.d., not determined. (**F**) DLD-1-vector and DLD-1-miR-21 cells were treated with various doses of VP-16 for 48 h. Whole-cell lysates were prepared and subjected to a Western blot analysis. (**G**) DLD-1-vector and DLD-1-miR-21 cells were treated with various doses of VP-16 for 24 h. Whole-cell lysates were prepared and subjected to a Western blot analysis. (**H**) DLD-1-vector and DLD-1-miR-21 cells were treated with various doses of VP-16 for 1 h. A band-depletion assay was performed as described in “Materials and Methods”.

**Figure 3 biomolecules-09-00467-f003:**
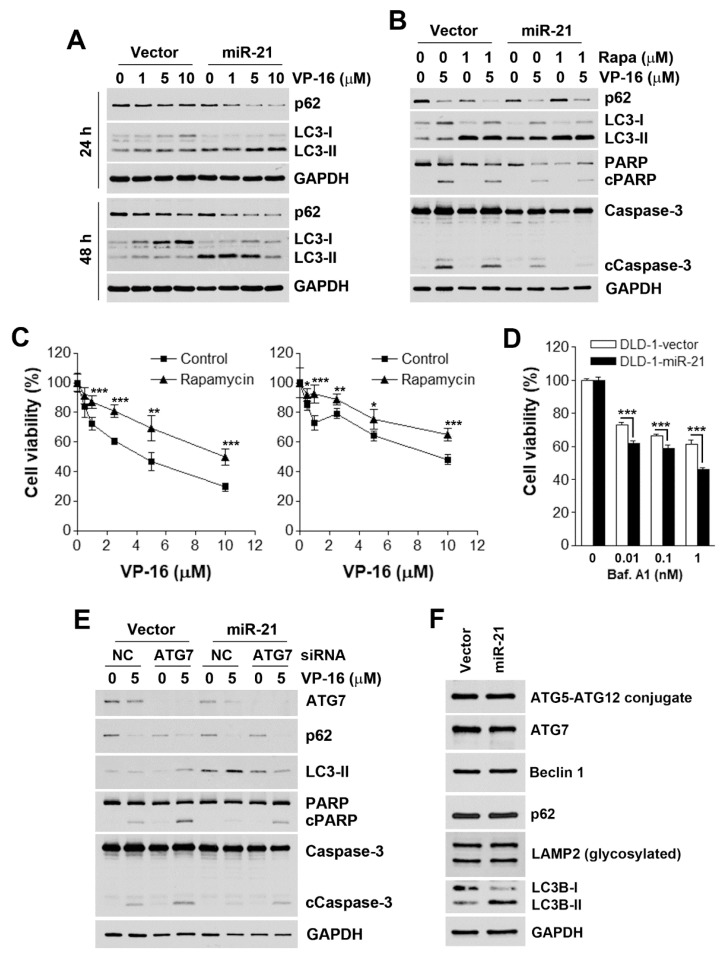
Effect of autophagy on miR-21-induced drug resistance. (**A**) DLD-1-vector and DLD-1-miR-21 cells were treated with various doses of VP-16 for 24 and 48 h. Whole-cell lysates were prepared and subjected to a Western blot analysis. (**B**) DLD-1-vector and DLD-1-miR-21 cells were treated with various doses of VP-16 with or without 1 μM rapamycin (Rapa) for 48 h. Whole-cell lysates were prepared and subjected to a Western blot analysis. (**C**) DLD-1-vector and DLD-1-miR-21 cells were treated with various doses of VP-16 with or without 0.5 μM rapamycin for 72 h. Cell viability was examined by an MTT assay. *p* < 0.05 (*), *p* < 0.01 (**), or *p* < 0.001 (***) indicates significant differences between rapamycin-treated and control cells. (**D**) DLD-1-vector and DLD-1-miR-21 cells were treated with various doses of bafilomycin A1 for 72 h. Cell viability was analyzed by an MTT assay. *p* < 0.001 (***) indicates significant differences between DLD-1-vector and DLD-1-miR-21 cells. (**E**) DLD-1-vector and DLD-1-miR-21 cells were transfected with ATG7 or non-targeting siRNAs for 48 h before exposure to 5 μM VP-16 for 48 h. Then, whole-cell lysates were prepared and subjected to a Western blot analysis. (**F**) The untreated whole-cell lysates from DLD-1-vector and DLD-1-miR-21 cells were analyzed by Western blot analysis.

**Figure 4 biomolecules-09-00467-f004:**
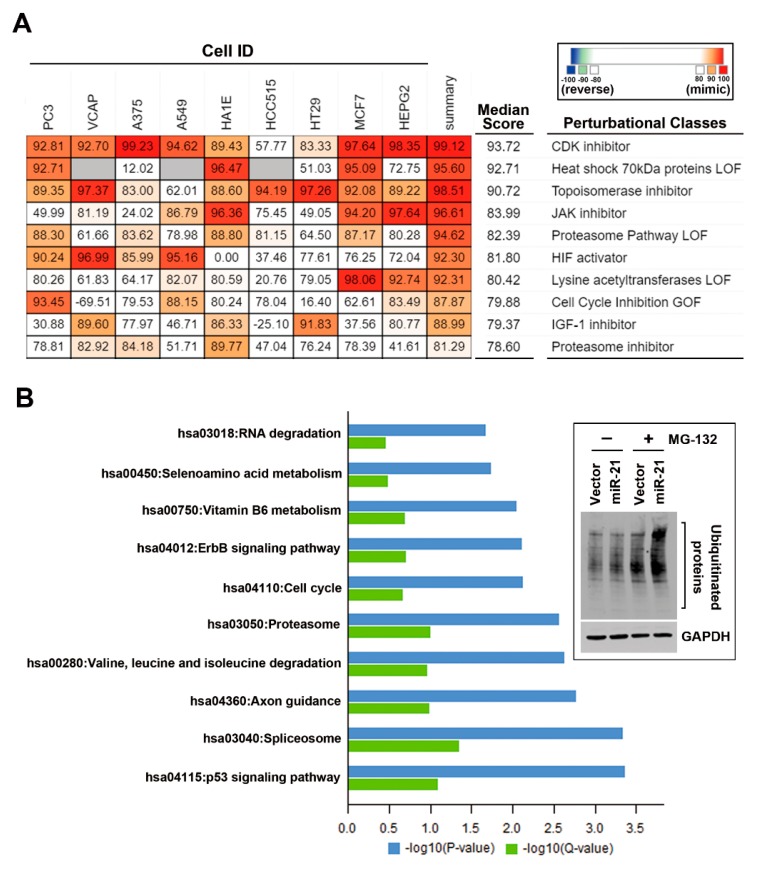
Microarray analysis of DLD-1-miR-21 cells. The total RNA isolated from DLD-1-vector and DLD-1-miR-21 cells was subjected to a microarray analysis as described in “Materials and Methods”. Differentially expressed genes (DEGs) were prepared for the next-generation Connectivity Map (CMap) analysis (**A**) and Kyoto Encyclopedia of Genes and Genomes (KEGG) pathway enrichment analysis (**B**). In (**A**), the top 10 most similar perturbational classes to miR-21 overexpression are shown. The gray grids indicate “not measured”. In (**B**), the top 10 enriched pathways are plotted on the Y-axis versus a measure of significance (negative logarithm of the *p* value or Q-value) on the X-axis. The Q-value was calculated by Benjamini. Embedded figure in (**B**): DLD-1-vector and DLD-1-miR-21 cells were treated with or without 10 μM MG132 for 4 h. Whole-cell lysates were prepared and subjected to Western blot analysis.

**Figure 5 biomolecules-09-00467-f005:**
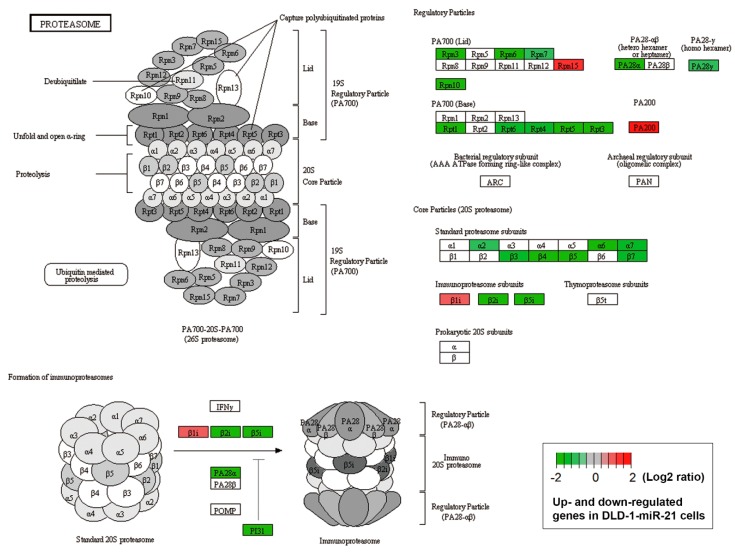
Proteasome pathway mapping of altered genes in DLD-1-miR-21 cells. A Kyoto Encyclopedia of Genes and Genomes (KEGG) pathway enrichment analysis was performed as described in “Materials and Methods”. Differentially expressed genes (DEGs) in DLD-1-miR-21 cells were mapped to the KEGG proteasome pathway (hsa03050). The genes highlighted in red and green indicated the upregulated and downregulated genes in DLD-1-miR-21 cells. The original data are shown in Appendix A.

**Figure 6 biomolecules-09-00467-f006:**
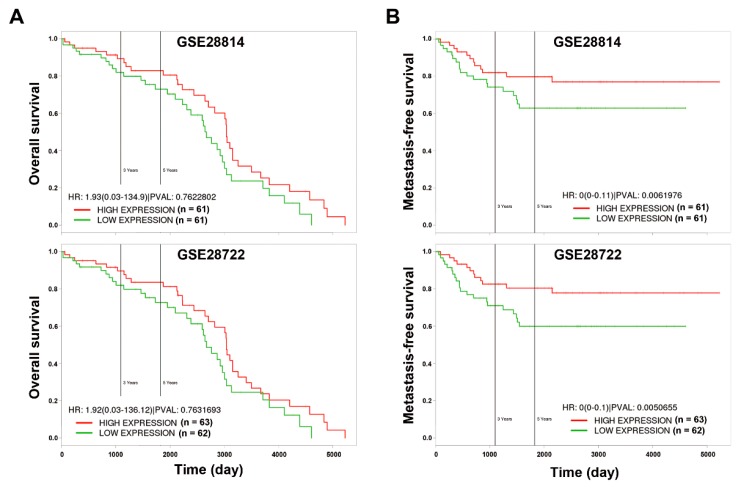
Prognostic impacts of proteasome gene alterations on overall (**A**) and metastasis-free (**B**) survival in colorectal cancer (CRC). The prognostic impact of the miR-21-downregulated proteasome gene signature (PSME3, PSMA1, PSMA2, PSMA3, PSMB2, PSMB3, PSMB4, PSMB5, PSMB8, PSMB10, PSMC2, PSMC3, PSMC4, PSMC5, PSMC6, PSMD3, PSMD4, PSMD11, PSME1, PSMF1, and PSMD6) in CRC was evaluated using the PROGgeneV2 database (http://www.compbio.iupui.edu/proggene/). Two CRC patient datasets (GSE28722 and GSE28814) were employed to generate the Kaplan–Meier survival plots. Patients with high (*n* = 61 or 63) and low (*n* = 61 or 62) miR-21 expression in GSE28722 and GSE28144, respectively, were bifurcated at the median value. HR, hazard ratio; PVAL, *p* value.

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
