# Peer review of "In Vitro and In Silico Mechanistic Insights into miR-21-5p-Mediated Topoisomerase Drug Resistance in Human Colorectal Cancer Cells"

_biomolecules, 2019, doi:10.3390/biom9090467_

Round 1

Reviewer 1 Report

Jung-Chien Chen and coworkers present an interesting key finding which they unfortunately surround by anless convincing convolute of incomplete data and far-fetched speculation. I would suggest to corroborate the hard core, remove the speculative stuff, and publish a concise version of tenable findings.

Key findings (the hard core):

The authors produce a CRC-cell line that overexpresses miR-21. They demonstrate that this manipulation significantly stimulates development of resistance to top2-poisons VP-16 and doxorubicine. It also slightly stimulates development of resistance to top1-poison SN-38, but that effect is insignificant. They further demonstrate that miR-21-overexpression enhances autophagic flux and downregulates some pathways in apoptosis induction. This hard core of the manuscript is potentially solid and possibly merits publication provided  the following questions and concerns could be clarified.

Concerns to be addressed:

Figure 2D: The observation that DLD-1-miR-21 exhibits a more cohesive growth pattern is potentially interesting. However the pictures shown in Fig. 2D are not convincing. If the authors wish to make this point they should perform more convincing tests. They could e.g. detach the cells in a protease-free manner (exposure to low Ca) and analyse particle size distribution by FACS.

Fig. 2G, why does Top 2a exhibit a double band in some of the lanes. Does this indicate effects of differential phosphorylation?

Fig. 2H: The quality of this figure is too poor to convincingly demonstrate band depletion effects. If the authors wish to argue in  terms of Topoiosmerase II activity , they should prepare nuclear extracts and measure kDNA decatenation activity.

Are any autophagy genes among the known targets of miR-21.

Ephemeral data and far-fetched conclusions (to be either corroborated or removed):

Putative involvement of the proteasome

The authors augment their above key findings with GEA possibly indicating down-regulation of certain proteasome components by miR-21. However, these findings are not corroborated up by proteasome activity assays or biochemical miR-21 target analyses, and , thus, remain somewhat ephemeral. The authors speculate in this context about compensatory upregulation of autophagic flux. This seems extremely far-fetched.

This whole part should either be worked out properly or removed from the manuscript.

The CRC connection

The authors start their investigation with a clinical argument, i.e. the enhanced expression of miR-21 in CRC and the poorer survival of CRC-patients exhibiting enhanced miR-21 expression. They correctly point out that standard chemotherapy for CRC consists of leucovorin, 5-FU and irinotecan (a topoisomerase I inhibitor). So miR-21 should affect response of CRC to exactly this drug combination. However, they challenge their miR-21 OE cell model in addition with topoisomerase II inhibitors VP16 and doxorubicin, which are not components of the above standard therapy and play no role in the clinical outcomes presented in Fig.1. Moreover, they find that miR-21-dependent differences in the development of resistance are significant only for the (clinically irrelevant?) Top2-trageted therapeutics. Subsequently, they go on talking summarily about effects of miR-21 on "topoisomerase-inhibitor resistance" while having demonstrated only a significant effect on "top2-inhibitor-resistance" but not topo1-inhibitor resistance. While their observation remains interesting, one wonders how relevant the findings are with respect to clinical therapy of CRC and whether it is heralded to discuss the data in this context. If they wish to maintain and strengthen the clinical quoin of their work, the authors should extend their in silico-investigations to tumor-entities that are actually treated with Top2-inhibitors and they should look whether miR-21 plays a role there. Otherwise, they should stick to the cell data and not construe a clinical context that is not supported by the data provided in this paper.

Along the same lines: How does the level of miR-21 overexpression in the cell – model compare to the level of miR-21 overexpression associated with poorer survival of the patients. This information could be incorporated into Figure 2 A (e.g. plot miR in poor-survival patiuemts normalized to the low-expressing patient group and OE in cell model normalized to control).

Critical survey of the conclusions:

Conclusion 1:  miR-21 overexpression induced drug resistance of CRC cells to topoisomerase inhibitors through inhibiting apoptosis and enhancing autophagic flux. This conclusion should be restricted to Top2-inhibitors but is otherwise valid.

Conclusion 2: miR-21 overexpression mimics the gene signature of topoisomerase inhibitors, which may lead to reduced responsiveness of CRC cells to drug treatment. The first part needs to be corroborated by wet work and the second part is pure speculation. Why should that gene signature have anything to do with drug resistance? Did any known rug-resistance signatures turn up in the GEA as well?

Conclusion 3: miR-21 overexpression inhibits the proteasome pathway through downregulating gene expressions, which may lead to enhancement of autophagic flux. Is there any experimental data or any report in the literature supporting this speculation?

Conclusion 4: … further exploration is needed. Very true: the authors should perform some more experiments to support some of their more far-reaching conclusions and claims.

Author Response

The responses to Reviewer 1 comments are not requested by the Academic Editor.

Reviewer 2 Report

The paper entitled “In vitro and in silico mechanistic insights into miR-21-5p-mediated topoisomerase drug resistance in human colorectal cancer cells” is well designed and easy to interpret.

The authors mast be check the grammar and the bibliography reported inside the text. (for example row 192). The quality of the images is appropriate but the graphs need more definition in particular Fig:1 and 5.

The conclusion is appropriate, the authors highlight that, miR-21 overexpression induced drug resistance to topoisomerase inhibitors in colorectal-cancer cell lines through inhibiting apoptosis and enhancing autophagy and the inhibition of the proteasome pathway

Reviewer 3 Report

The manuscript, biomolecules-576173, provides an insight into the mechanism of miR-21-5p-mediated resistance to (anti-cancer) topoisomerase inhibitors.

Firstly, I would suggest to the editor to consider relevance of the paper to the journal, Biomolecules, as personally, I believe this would be much better suited for submission to a cancer-specific journal, such as Cancers.

For review, I would suggest that the authors address the following points:

Rewrite the abstract to avoid repetition, as the results section is repeated. The authors should consider additional methods to validate changes in cell viability. MTT assays are not able to discriminate between cytostatic and truly cytotoxic responses. Consider more comprehensive measures of loss of cell membrane integrity such as LDH assays, and apoptosis measurements via flow cytometry. Results section, lines 192/193 have duplication of references. With regard to cell viablility assays (Figures 2E), for 5-FU, SN-38, and doxorubicin only minor changes are observed between the vector control or miR-21 treated cells are evident. For the latter two, these may well be significant, but only at relatively high concentrations? The authors should mark on each data point whether there is significance. Secondly, by extending the data points an IC50 value could be generated for each condition, enabling comparison with other studies. Furthermore, the authors should provide relevant details that support the use of the concentrations of agents, what concentrations would be used in vivo? It is pertinent that the second set of experiments (Figure 3) only used VP-16 and at relatively low concentrations for which there was a significant difference between the vector control and miR-21 treated cells. Presumably, this was not undertaken for the other agents SN-38 and doxorubicin?

Round 2

Reviewer 1 Report

The authors have made some textual alterations and attenuated some of their bolder conclusions. Otherwise they seem indifferent to my concerns and suggestions. I could not find any new data, but I may have missed that point since a point-by-point reply to my concerns was not provided or not transmitted, rendering it difficult to analyse the revision properly.

The paper provides a host of interesting data and observations

It is demonstrated that a cell line subjected to miR-21 overexpression

  1.) becomes resistant to SN38, VP16 and doxorubicin

  2.) increases autophagic flux (possibly related to effect 1 given the rapamycin and bafilamoycin effects)

 3.) upregulates proteasome genes 

Coincidentally, CRC-patients have a poorer survival

  4.) when miR-21 -expression is increased

  5.) when expression of certain proteasome genes is increased

In summary these correlations suggest the speculative conclusions drawn by the authors, namely that the above effects 1-3 seen in a manipulated cell line are interrelated and provide the explanation for the phenomena 4 and 5 seen in CRC therapy response. A causal relationship and mechanistic explanation of the so far circumstantial evidence is lacking.

In summary, this is a potentially very interesting but so far rather preliminary study.